# Factors Associated with Dietary Patterns of Schoolchildren: A Systematic Review

**DOI:** 10.3390/nu15112450

**Published:** 2023-05-24

**Authors:** Laura Rangel Drummond de Menezes, Rafaela Cristina Vieira e Souza, Pollyanna Costa Cardoso, Luana Caroline dos Santos

**Affiliations:** 1Nutrition Department, Universidade Federal de Minas Gerais, Belo Horizonte 30130-100, MG, Brazil; lauramenezes17@gmail.com (L.R.D.d.M.); luanacstos@gmail.com (L.C.d.S.); 2Nutrition Department, Universidade Federal de Juiz de Fora, Governador Valadares 35010-180, MG, Brazil; pccnutri@gmail.com

**Keywords:** processed foods, food intake, food consumption, children

## Abstract

The evaluation of food consumption in childhood is essential to help understand the effect of food choices on health. The objective of this study was to conduct a systematic review of studies that identified the dietary patterns in schoolchildren (7–10 years old) and their associated factors. Observational studies published in the last ten years were searched in the databases BVS (Virtual Health Library), Embase, PubMed, Scopus, and Web of Science. The Newcastle Ottawa Scale was adopted to evaluate the articles’ quality. The studies covered schoolchildren, children, and adolescents as part of the sample. We selected 16 studies, 75% of which were considered good/very good and seven mentioned three food patterns. A dietary pattern considered unhealthy was identified in 93.75% of the studies, having as associated factors to its consumption: higher screen time, low bone mass, gain of weight and fat in children, and meal skipping. The children who usually had breakfast showed greater adherence to the dietary pattern consisting of healthier foods. The children’s dietary patterns were related to their behavior, nutritional status, and family environment habits. Food and nutrition education’s effective actions, as well as the regularization of the marketing of ultra-processed foods, must be stimulated and inserted in public policies as a way to promote and protect children’s health.

## 1. Introduction

The eating scene in recent decades points to an increase in the consumption of ultra-processed foods, which are high in energy, fat, sugar, and salt, opposed to the consumption of fruits and vegetables and, therefore, a reduction in vitamins, minerals, and fibers [1,2,3,4,5,6,7]. These facts contributed to the increase in childhood overweight and obesity, which are early risk factors for the development of cardiometabolic complications, diabetes, and cancer [8]. It is estimated worldwide that 11% of children aged 5 to 9 years have obesity, which corresponds to 72 million children [9]. In Brazil, 14.37% of children aged 7 to 10 years are overweight for their age [10].

The assessment of food consumption in childhood is essential to help to understand the effect of food choices on health in the short term and even during adult life, as eating habits acquired in childhood can last in the long term [11]. This evaluation can be carried out using knowledge of dietary patterns, which were highlighted in recent years. Dietary patterns are defined as the set or group of foods consumed by a certain population obtained through statistical methods of aggregation or reduction in components. They can provide a global vision of the diet and that helps with the early identification of dietary deviations and the use of appropriate interventions [12,13].

Studies from the last decade with different populations identified several dietary patterns, such as the “Western”, “Mediterranean”, “traditional”, and the “healthy” one, referring to the consumption of food groups from each country or region. Each of these patterns presents specific associations with lifestyle characteristics, thus presenting diverse health effects [14,15]. Because dietary patterns are inherent in certain communities and vary according to sex, age, culture, ethnicity, socioeconomic status, and food availability, it is important that we analyze them in specific groups [16]. Data on children’s dietary patterns are still scarce, in view of the concentration of research on the topic among adults [17,18].

Childhood is recognized as a fundamental stage for forming and establishing eating habits. Psychological, socioeconomic, and cultural aspects are potential determinants in the choice of food during this period, interfering with the choices and risk behaviors linked to the current epidemic of childhood obesity and chronic diseases [17]. Therefore, it is relevant to identify inadequate eating habits in this stage of life in order to reflect and intervene in favor of health promotion and reduction in diseases throughout life. Considering the gap in the literature on the subject, especially for children aged 7 to 10 years old, the authors conducted a systematic review of studies that identified the dietary patterns in schoolchildren and evaluated their associated factors.

## 2. Materials and Methods

### 2.1. Design

This review was registered in the International Prospective Register of Systematic Reviews (PROSPERO Network, nº 399923—https://www.crd.york.ac.uk/prospero/display_record.php?ID=CRD42023399923, accessed on 20 April 2023). In line with the development of written production, the PRISMA instrument (Preferred Reporting Items for Systematic Reviews and Meta-Analyses) was used. All stages were independently conducted by three researchers (LRDM, PCC, and RCVS). In case of disagreement, a fourth evaluator (LCS) was consulted to reach a consensus.

### 2.2. Databases Search

A search for scientific articles was conducted until February 2023 in the electronic databases BVS, Embase, PubMed, Scopus, and Web of Science. The descriptors DeCS/MeSH and free terms were selected as well as their translations and synonyms: “food consumption”, “dietary patterns”, “food intake pattern”, “child”, “child nutrition”, “influencing factors”, and “influence” (Appendix A). 

### 2.3. Eligibility Criteria

All papers in Portuguese, English, and Spanish, published from 2010 to February 2023, were included in this study. The PECOS strategy was used to search the articles, using as elements: population: schoolchildren aged between 7 and 10 years; exposition: associated factors; comparator: not applicable; outcome: dietary pattern; and study design: cross-sectional and cohort. The exclusion criteria adopted were studies of reviews, meta-analyses, clinical trials, and observational studies conducted with different groups: pregnant women, nursing mothers, nursing infants, preschoolers, adults, and the elderly.

After removing duplicates, the articles were screened by reading the titles, abstracts, and keywords, using the reference manager EndNote Web^®^.

### 2.4. Assessment of the Individual Quality of the Articles

The evaluation of the methodological quality of the eligible articles was carried out in duplicate with the help of the Newcastle Ottawa scale (NOS) for cohort studies [18] and adapted for cross-sectional studies, contemplating seven items divided into three domains: selection, comparability, and outcome. The interpretation of the NOS score was distinguished in very good (9–10 points), good (7–8 points), satisfactory (5–6 points), and unsatisfactory (0–4 points) studies. Articles evaluated as unsatisfactory were excluded from this review; however, such a classification was not present among the eligible articles.

### 2.5. Classification of the Dietary Patterns from the Selected Articles

Ultra-processed foods are industrial formulations entirely, or are mostly produced from substances extracted from food (oils, fats, sugar, starch, proteins), derived from food constituents (hydrogenated fats, modified starch), or synthesized in the laboratory to provide products with attractive sensory properties. They are ready-to-eat or ready-to-heat foods; therefore, little or no culinary preparation is required, making them affordable and convenient. They are usually combined with a sophisticated use of additives to make them durable and hyper-palatable. However, they have very low nutritional quality and tend to limit the intake of in natura or minimally processed [19] foods.

Thus, among the articles considered eligible for this review, those whose dietary pattern was predominantly composed of ultra-processed foods [19] were classified as “unhealthy” and the rest as “healthy”.

## 3. Results

A total of 2747 articles were identified, and 716 out of them were excluded due to duplicity. After reading titles, abstracts, and keywords, 37 studies were selected for a full reading. 16 out of these were eligible for this review (Figure 1).

### 3.1. General Characteristics of the Included Studies

Table 1 presents the study design, population characteristics, methods for assessing food consumption, and dietary patterns of schoolchildren, associated factors as well as the quality of the studies.

Of the 16 articles included, 9 studies were cross-sectional and 7 longitudinal. The studies were conducted in Asia (n = 4), Central America (n = 4), South America (n = 3), Europe (n = 3), and Oceania (n = 2) [20,21,22,23,24,25,26,27,28,29,30,31,32,33,34,35]. Eight studies evaluated only children [21,23,26,30,32,33,34,35] and eight evaluated children and adolescents [20,22,24,25,28,29,31]. Studies with adolescents were included because they also addressed schoolchildren. The most used method for assessing food consumption was the food frequency questionnaire (FFQ) used in ten of the studies [22,23,25,26,28,30,31,33,34,35], followed by 24 h recall (n = 3) [20,27,29] and food record (n = 3) [21,24,32].

Regarding the quality of the studies, a mean of 7.53 (SD = 0.63) was identified, with 75% considered good and very good (n = 12). The main failures observed were self-reported results (100%) and not adjusted for confounding variables (such as gender, age, and/or socioeconomic status) (31.25%) [23,24,27,31,32].

### 3.2. General Characteristics of Dietary Patterns of the Included Studies

The number of dietary patterns obtained in each study varied. The majority of the studies (seven studies) found three patterns [22,23,24,26,30,32,34], four identified four patterns [20,27,28,33], three studies found five patterns [21,25,35], and two identified two patterns [29,31]. The majority (93.75%) identified a dietary pattern considered as “predominantly unhealthy”, that is, patterns that are made up predominantly of unhealthy foods, being called “snacks (n = 4), “junk convenient” (n = 2), “unhealthy” (n = 1),” less healthy” (n = 1), “industrialized” (n = 1),” saturated fat” (n = 1), “junk food” (n = 1), “fast food and fried food” (n = 1), “obesogenic” (n = 1), “snacks” (n = 1), “soft drinks” (n = 1), “canned soups and meals” (n = 1), “sweetened drinks” (n = 1), “modern” (n = 1), “milk and chocolates” (n = 1), “high fat” (n = 1), “high sugar” (n = 1), “Western” (n = 1), “diet” (n = 1), “refined cereals and animal organs” (n = 1). In contrast, dietary patterns classified as “predominantly healthy” (75%) were identified in the studies as “typical food” (n = 4), “healthy” (n = 3), “Mediterranean” (n = 1), “vegetables” (n = 1) and other “diverse” (n = 1), “legumes, whole grains and red meat” (n = 1), “fruit, milk and egg” (n = 1), “shellfish, mushroom and eggs” (n = 1), “beverages, fish and low-fat milk” (n = 1).

**Table 1 nutrients-15-02450-t001:** Characteristics of the included studies that evaluated the factors associated with the eating patterns of schoolchildren.

Authors; Year; Country	Design	Method/Sample	Dietary Assessment	Dietary Patterns and Composition	Associated Factors	PointsNOS
Liao et al. [35]; 2022; China	Cross-sectional	465 children from 6 to 9 years old	FFQ	5 patterns: Legumes, whole grains, and red meat: fresh vegetables, whole grains, and red meat. Milk, fruit and eggs: fruits, whole milk, and eggs. Shellfish, mushrooms, and nuts: molluscs and Shellfish, mushrooms, and nuts. Drinks, fish, and low-fat milk: water and sea fish, low-fat milk, soup, and drinks. Animal organs and refined cereals: refined cereals, canned vegetables, red meat, and animal organs.	“Milk, fruit, eggs” pattern and better bone density. “Animal organs and refined cereals refined” pattern and low bone density.	7
Lee et al. [23]; 2017; China	Cohort	154 children aged 7 and 9 years old	FFQ	3 patterns: Healthy: vegetables and beans. Animal source: meats and fish. Snacks: sweets, soft drinks, and breads.	“Animal source” pattern and female. More TV time was and the consumption of the “snacks” pattern. Habit of breakfast and the regularity of a varied diet and greater adherence to the predominant “healthy” dietary pattern.	7
Li et al. [25]; 2016; China	Cohort	283 pairs of twins aged 7 and 15 years old	FFQ	5 patterns: Vegetables and fruits: beans and fruits. Meat: red meat, fish, eggs, and dairy products. Beverages: soft drinks and juices. Snacks: snacks, nuts, sweets, and chocolates. Fried and fast foods: Western fast food and Chinese fried food.	Higher levels of maternal education and consumption of “Fried and fast food” and lower heritability for the pattern “vegetables”.	9
Choi et al. [32]; 2011; Korea	Cohort	284 children aged 7 to 8 years	3 food records	3 patterns: Korean: seasonings, vegetables, white rice, and kimchi. Modified Western: kimchi, beverages, and potatoes. Western: noodles, ramen, cookies, crackers, chips, sugar, sweets, pizza, and hamburgers.	No association	6
García-Chávez et al. [20]; 2020; Mexico	Cross-sectional	2751 children aged 5 to 11 years	24 h recall	4 patterns: Traditional: tortilla, legumes, egg, sugar-sweetened beverages, bread, and other cereals. Industrialized: milk drinks with sugar, snacks made from flour, corn, or potato, fast food, desserts, pastries and sweets, and industrialized beverages. Varied: tortilla or corn dough, cereals with sugar, meat, and sausages, dairy drinks, fruits, rice, and pasta food groups. Modern: tortas and sandwich, and breakfast cereal with sugar food groups.	“Modern” pattern and obesity.	7
Zamora-Gasga et al. [24]; 2017; Mexico	Cross-sectional	724 children from 9 to 12 years old	2 food records	3 patterns: Modified Mexican Diet: meats, oils, cereals, snacks, and drinks. Traditional Mexican Diet: legumes, vegetables, snacks, sauces, and seasoning. Alternative Mexican Diet: fish and vegetables.	Inverse association between the “traditional” pattern and weight and body mass index.	7
Galvan-Portillo et al. [22]; 2018; Mexico	Cross-sectional	857 children from 5 to 15 years old	FFQ	3 patterns: Diverse: other fruits, other vegetables, melons, citrus fruits, orange vegetables, fish, probiotic dairy products, dry beans and peas, berries, darkgreen leafy vegetables and starchy vegetables. High fat pattern: processed and high-fat foods, refined grains, processed meat, red meat, whole grain products, fats, and processed dairy. High sugar pattern: grain products with more fat and sugar, sugar-sweetened beverages, and high-fat and sugar milk products.	Higher level of parental education lower adherence and “high sugar pattern’. Instability in the mother’s occupation and the “high fat” pattern. Screen time greater than 1.25 h/day and the “high sugar content” dietary pattern.	7
Vieira et al. [21]; 2019; Brazil	Cohort	403 children from 4 to 7 years old	3 non-consecutive food records	5 patterns: Traditional: white rice, beans, vegetables, tubers, polenta and flours, meat, fish, and eggs. Unhealthy: foods/food groups with high sugar and fat content, such as: artificial juice and soft drinks, fried foods, snacks and sausages, sweets, and stuffed cookies. Milk and chocolate: milk and dairy products, and chocolate and sugar. Snack: typical bakery foods, such as: breads, cakes and cookies, butter and margarine, and coffee and tea. Healthy: natural juices, fruits, vegetables, broths, and soups.	No association	7
Silva et al. [31]; 2012; Brazil	Cross-sectional	1136 students from 7 to 14 years old	FFQ	2 patterns: Obesogenic: foods that are sources of fat in general and saturated and high-glycemic fats. Traditional: meat and derivatives, eggs, cereals and derivatives, vegetables and fruits.	Lowest maternal educational level was negatively associated and “obesogenic” pattern. Inverse association between maternal income and “obesogenic” pattern.	7
Shroff et al. [28], 2013; Colombia	Cohort	961 children from 5 to 12 years old	FFQ	4 patterns: Snacking: candy, ice cream, packed fried snacks, soda and sugar-sweetened fruit-flavoured drinks. Cheaper protein: NM Traditional/starch: NM Animal protein: NM	“Snacks” pattern and greater gain in Body Mass Index/year and the ratio of subscapular and triceps skinfolds.	7
Kaiser et al. [34]; 2015; USA	Cross-sectional	217 children from 2 to 8 years old	FFQ	3 patterns: Fast/convenience: pizza, hamburgers, hot dogs, canned fruit, instant noodles, spaghetti sauce, fried potatoes, chips, and soft drinks. Vegetables: vegetables in different forms, including vegetable soup, other cooked vegetables (except potatoes), lettuce or cabbage, other raw forms of produce. WIC: low-fat milk, fresh fruit, ready-to-eat cereal, juice, and corn tortillas.	No association	6
Rodrigues et al. [26], 2016; Portugal	Cross-sectional	1063 children aged 6 to 8 years	FFQ	3 patterns: Portuguese diet: meat, pasta, potatoes, and rice. Mediterranean diet: fruits, vegetables, legumes, and fish. Saturated fat diet: fast food, eggs, and butter.	Higher socioeconomic status and the pattern “Portuguese diet” and “Mediterranean diet” and inverse and “saturated fat diet” pattern. Males and higher consumption of the “Portuguese diet” pattern. Habit of watching TV and the “saturated fat diet” pattern.	7
O’Brien et al. [29], 2013; Ireland	Cross-sectional	483 children from 7 to 13 years old	Usual 3-day food record	2 patterns: More healthful: rice, pasta, starches, and grains, wholemeal bread and wholemeal baked goods, breakfast cereals low-fat milk, potatoes, vegetables, fruits, fish and fish products, and poultry and poultry dishes. Less healthful: white bread and white bread products, butters and spreads, chips and processed potato products, processed red meats, sugars and confectionery, and high-calorie beverages.	Families with a high socioeconomic level and a “healthier” pattern.	5
Rahmawaty et al. [27]; 2014; Australian	Cohort	1110 children from 9 to 13 years old	2-day 24 h recall	4 patterns: For boys Snacks foods: processed snack foods, chicken (all types, including nuggets, curry), sandwiches, and breakfast cereals, with low intakes (negative factor loadings) of fish, eggs, and fatty meats Soft drinks: soft drink and take-away items (meat, fish, cereal, and mixed vegetables), with low consumption of fish. Vegetables: high consumption of vegetables, nuts and seeds, and sweet potatoes. Pork and meat chops, steak, and mince: high consumption of pork and meat chops, steak and mince, cheese, potatoes, ice cream, and soft drinks, with low consumption of fish and rare intake of eggs. For girls Vegetables: high consumption of vegetables, potatoes, sausages, nuts and seeds, sweet potatoes, in addition to soft drinks, with low consumption of fish and rare consumption of yogurt. Take-aways: take-aways (meat, fish, cereal, and mixed vegetables), specialty (meat, cereal, and mixed vegetables), soft drinks, and snack foods, with low consumption of fish and eggs. Coffee, iced coffee drinks: high consumption of tea, coffee, iced coffee drinks and vegetable mix dishes, with low intake of fish and yogurt. Canned meals and soup: high consumption of soup and canned meals and soup, and special drinks (such as energy, sports, protein-based, malt, hydroelectrolytic, with probiotics), low consumption of yogurt and eggs.	No association	6
Wall et al. [30]; 2013; New Zeland	Cohort	550 children aged 3.5 years and 591 children aged 7 years	FFQ	3 patterns: Tradicional: cauliflower, peas, mixed vegetables, potatoes, pumpkin and beef as the main dish. Junk: candy bars, hamburgers, soft drinks, chips, chocolate and lollies. Healthy: pineapple, tomatoes, cucumber, ‘other’ green vegetables, celery, mixed grain bread and a negative weighting for white bread.	Male and the “traditional” pattern.	7
Oellingrath et al. [33], 2010; Norway	Cross-sectional	941 children aged 9 to 10 years	FFQ	4 patterns: Snacking: snack items and sugar-sweetened drinks consumed between meals, combined with low breakfast and dinner frequency and low intake of water, vegetables and brown bread. Junk/convenient: high-fat and high-sugar processed fast foods such as French fries, processed pizza, processed meat products, sweets, ice cream and soft drinks. Varied Norwegian: food items typical of a traditional Norwegian diet, such as fish and meat for dinner, brown bread, regular white or brown cheese, lean meat, fish spread, and fruit and vegetables. Dieting: artificially sweetened soft drinks, low fat cheese and fat- and sugar-reduced yoghurt, and was negatively associated with sugar-sweetened soft drinks.	Low maternal educational level and higher scores of the “snacking” pattern. Male and higher scores for the patterns “snacking” and “junk/convenient”. “Varied Norwegian” or “diet” dietary pattern and overweight. Paternal and maternal overweight and higher “diet” pattern scores Maternal overweight and higher “snacking” scores.	7

### 3.3. Factors Associated with Dietary Patterns

Table 2 represents the number of associations found in the articles, between all dietary patterns and associated factors found in each study in a broad but objective way. Associations were demonstrated directly or inversely, with patterns that were classified as healthy or unhealthy. In each study, it was possible to find more than one association between dietary pattern and associated factor.

### 3.4. Socioeconomic Characteristics

Galvan-Portilho et al., evaluated 857 Mexican children and adolescents aged 5 to 15 years and identified that the higher level of parental education showed a lower probability of adherence to the “high sugar” standard (grain products with more fat and sugar, sweetened beverages and dairy products high in fat and sugar, few fruits and vegetables) [22]. Li et al., found among 283 pairs of Chinese twins, aged 7 to 15 years, that higher levels of maternal education were associated with greater heritability for consumption of “fast food and fried foods” (Western fast food and Chinese fried foods) and lower heritability for the pattern “vegetables” (vegetables, beans, and fruits) [25]. In comparison, the study conducted by Oellingrath et al., with 924 Norwegian children aged 9 to 10 years identified that the low maternal educational level was associated with higher scores of the “snack” pattern (snacks and sweetened drinks consumed between meals, low frequency of breakfast and dinner, and low consumption of water, vegetables, and whole grain bread) [33]. In a study with 1.136 Brazilian children aged 7 to 14 years, Silva et al., observed that the lowest maternal educational level was negatively associated with the “obesogenic” pattern (foods source of fats in general and saturated fats and high glycemic index) [31].

The study by O’Brien et al., carried out with 483 Irish children aged 7 to 13 years, found an association between families with a high socioeconomic level (compared to low and medium levels) and a “healthier” pattern (higher intake of food groups: “rice, pasta, starches, and grains”, “wholemeal bread and pastries”, “breakfast cereals”, “skimmed milk”, “potatoes”, “vegetables”, “fruits”, “fish and fish products” and “poultry and poultry preparations”) [29]. Rodrigues et al., in their study with 1.063 Portuguese children aged 6 to 8 years, observed an association between higher socioeconomic status (parental education and occupation) with the pattern “Portuguese diet” (rice, pasta, meat, and potatoes) and “Mediterranean diet” (vegetables, fruits, and fish), and negative association with the pattern “saturated fat diet” (fast food, eggs, and butter) [26].

Regarding income, Silva et al., reported an inverse association between maternal income and the percentiles of the distribution of the “obesogenic” pattern [31]. Li et al., in a study with twins, detected an association between higher family income and adherence to the “meat”, “fast food and fried foods” pattern [25]. In the study by Galvan-Portilho et al., it was observed that instability in the mother’s occupation (unemployed or informal work) was associated with a greater probability of following the “high fat” pattern (processed and high-fat foods, refined grains, processed meats, red meats, whole grain products, processed dairy products, Mexican dishes) [22].

There were no consensual results when comparing sexes. Rodrigues et al., found that males were associated with a higher consumption of the “Portuguese diet” pattern, configured as a healthy diet [26]. Similarly, Wall et al., identified an association between the male gender with the “traditional” pattern (cauliflower, peas, vegetable mix, potatoes, pumpkin, and meat as the main dish) among 591 New Zealand children aged 7 years [30]. Oellingrath et al., observed that boys had higher scores for the patterns “snack” and “junk/convenient” (fast food processed with high fat and sugar content, such as French fries, industrialized pizza, processed meats, sweets, ice cream, and soft drinks) [33]. In the study by Lee et al., with 154 Korean children at 7 and at 9 years of age, there was an association between the “animal source” pattern (meat and fish), considered healthy, and being female [23].

### 3.5. Screen Time

Screen time was associated with the “predominant unhealthy” pattern among schoolchildren in the included studies. Lee et al., observed that having more TV time was associated with the consumption of the “snacks” pattern (sweets, soft drinks, and bread) [23]. Similarly, Rodrigues et al., identified an association between the habit of watching TV with the “diet based on saturated fat” pattern [26]. In the study by Galvan-Portilho et al., it was found that screen time higher than 1.25 h/day was associated with a higher probability of the child having a “high sugar” dietary pattern [22].

### 3.6. Nutritional Status

The included studies showed an association between nutritional status and the dietary pattern, and they showed higher weight gain/BMI or adiposity in schoolchildren with higher adherence to unhealthy dietary patterns. In 2751 Mexican schoolchildren (aged 5 to 11 years), the “modern” pattern (pies and sandwiches, and breakfast cereal with sugary groups) was associated with obesity [20]. Shroff et al., observed, in their study with 961 Colombian children aged 5 to 12 years, that those in the highest quartile of adherence to the “snacks” pattern (foods with high energy density and low nutrient density: sweets, ice cream, fried snacks, soft drinks, and sweetened drinks with fruit flavor) had a greater gain in BMI/year and the ratio of subscapular and tricipital skinfolds compared to children in the lower quartile [28].

In contrast, Zamora-Gasga et al., evaluated 724 Mexican children aged 9 to 12 years and observed a negative association between the pattern classified as “traditional” (large intake of legumes, vegetables, snacks, sauces, and seasoning) and weight and body mass index [24]. Meanwhile, in a study with 9 and 10-year-olds who had a “varied Norwegian” or “dieting” dietary pattern, the schoolchildren were much more likely to be overweight [33].

### 3.7. Other Factors

The habit of breakfast and the regularity of a varied diet were associated with greater adherence to the predominant “healthy” dietary pattern in the study by Lee et al. [20] conducted with 154 children aged 7 to 9 years. A study with children aged 2 to 8 years [34] found an association between meal skipping and adherence to the pattern “fast/convenient” considered mostly unhealthy.

The nutritional status of the parents was mentioned as associated with the dietary pattern of the schoolchild. Oellingrath et al., found that paternal and maternal overweight were associated with higher “diet” pattern scores and maternal overweight with higher “snack” scores [33].

Regarding bone development, Liao X et al., associated the “fruit–milk–eggs” pattern, which is considered a healthy pattern, with better bone density. In contrast, the pattern “refined cereals and animals’ organs”, considered unhealthy, was associated with low bone density [35].

## 4. Discussion

The findings of this review pointed to a wide diversity in the identification of dietary patterns of schoolchildren, as well as among the associated factors analyzed by the articles. One finding observed was the connection between adherence to a pattern considered mostly unhealthy and the habit of eating while watching TV, weight gain in schoolchildren, and meal skipping. On the other hand, predominantly healthy patterns were more frequent among schoolchildren with the habit of eating breakfast.

The association between socioeconomic factors and the patterns was inconclusive, regardless of the country under study. Some hypotheses were postulated to explain the divergences. One of them was the specificity of the population regarding the type of dietary pattern consumed, as they can vary according to the various determinants. Additionally, the socioeconomic characteristics of each region make comparisons difficult and denote the complexity of the interaction between these factors and food consumption [36].

It is worth mentioning that, regarding screen time, the investigation was consistent with the evidence. Schoolchildren were found to snack more while watching television; therefore, the longer the screen time, the higher the consumption of ultra-processed foods (sweets, soft drinks, and chips) and the lower consumption of fruits and vegetables [37].

The literature recently observed an average exposure of 7 h of electronic media per day in young people aged 8 to 18 years in the USA; as a result, food consumed in front of the television represents about 20 to 25% of the daily energy intake of this public [38,39]. Another aspect that deserves to be highlighted is the marketing that comes from screen time. One study reported that children with access to unhealthy food and beverage marketing increased dietary intake and preference for foods and beverages with high energy density and imbalanced nutrition. Marketing of unhealthy foods and beverages increased food intake and influenced food preference in children during or shortly after exposure to advertisements [40].

In line with this hypothesis, the authors indicated that when watching television or using a computer, schoolchildren usually tend to eat unhealthy foods because they are practical and palatable. In this sense, the time spent with physical activities is also associated with typical traditional foods (meats, including fish, wholemeal bread, fruits, and vegetables) and conversely in the habit of consuming low-fat and low-sugar foods [41,42].

Regarding nutritional status, this study’s findings are consistent with the assumptions that a healthy diet—higher consumption of in natura and minimally processed food as opposed to ready-to-eat foods and ultra-processed beverages, low nutritional, and high energy density foods—can impact the nutritional status, food consumption, and eating behaviors in childhood and, consequently, health promotion [5,6,10].

The typical nutritional composition of ultra-processed foods, present in most patterns characterized as predominantly “unhealthy”, can contribute to inadequate micronutrient consumption among schoolchildren and impair child growth and development, in addition to increasing the risk of diseases and non-communicable illnesses such as obesity and cardiometabolic alterations [43,44].

Regarding the relationship between patterns and bone mass, the deposition of this tissue occurs early in life, particularly during childhood and puberty, and the literature refers to them as crucial periods for health and the subsequent appearance of osteoporosis. Although genetic factors contribute to the peak of bone mass, adequate nutrition, with a balanced composition of key nutrients for bone formation such as calcium and vitamin D, is essential for the modulation of osteogenesis [45,46,47,48].

In the context of parental weight and food consumption, parental overweight was associated in one study with a low frequency of breakfast and dinner and excessive consumption of snacks. These results corroborate the family food environment as a strong predictor of children’s food practices and preferences [48,49].

A direct relationship was found between a “predominant healthy pattern” and the habit of eating breakfast in the study by Lee et al., with 380 Korean children [23]. Such a result was coherent with the one found by Kaiser et al. [34] in a study conducted with 217 immigrant parents and their children in rural communities. It is important to emphasize the relevance of breakfast as a meal that provides nutrients and energy, and its omission can impair school performance, satiety, and weight control, besides favoring the desire of eating caloric snacks. Furthermore, the consumption of breakfast seemed to be associated with eating fewer meals during the day; thus, it is a factor of protection against child obesity [50,51].

Some limitations should be considered in this review regarding the diversity of methods to evaluate food consumption between studies, making it difficult to compare the findings, ‘the different assessments of food intake can impair the knowledge of the foods consumed and, thus, can lead to different results of dietary patterns” [2]. However, it should be noted that this is one of the few comprehensive investigations carried out that contemplated dietary patterns and associated factors in this age group, which lacks results and, at the same time, is sensitive to the effects of food consumption. The data obtained can contribute to the design of preventive and protective measures for children’s health.

## 5. Conclusions

Schoolchildren with the habit of having breakfast and those with regular consumption of a varied diet showed a more frequent adherence to the patterns composed of healthy foods. Alternatively, longer screen time, overweight schoolchildren, higher adiposity in parents, and meal skipping were associated with dietary patterns composed of unhealthy foods and unhealthy habits.

Given these findings, it is essential that public policies aimed at food and nutrition education actions in schools and in the community are implemented and intensified. The care and attention of health professionals is necessary in promoting and monitoring inadequate eating habits in childhood, as early nutritional intervention can favor the prevention of diseases in the near and long future. In addition, it is essential to ensure healthier environments that protect from the exposure of ultra-processed food marketing, to regulate the commercialization of these foods, to propose regulatory actions for advertisements and to encourage the consumption of in natura foods and to inhibit free access to ultra-processed foods, facilitating the understanding of labels and overcoming barriers to promote healthy eating.

## Figures and Tables

**Figure 1 nutrients-15-02450-f001:**
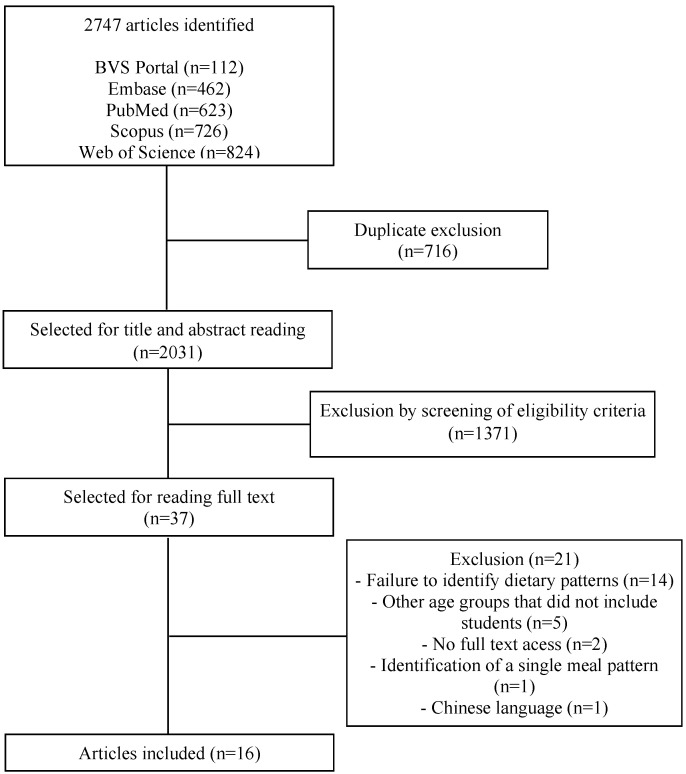
Flowchart of the process of selecting studies from the systematic review on dietary patterns and factors associated with food consumption in schoolchildren.

**Table 2 nutrients-15-02450-t002:** Summary of associations (direct or inverse) found in articles that assess factors associated with eating patterns of schoolchildren.

	Patterns			
	Healthy		Unhealthy	
Associated Factors	Direct	Inverse	Direct	Inverse
Economic situation	3			1
Parents’ education		1	1	3
Per capita income	1	1	1	1
Maternal occupation			2	
Parents’ overweight			2	
Child’s age			1	
Child’s gender	3		2	
Obesity/Adiposity of the child		1	3	
Screen time			3	
Habit of eating breakfast and variety in food	1			
Missing meals from the child	1		1	
Bone mineral density	1			1

Note: Unhealthy pattern was called fast food/convenience pattern [34], modern pattern [20], unhealthy pattern, milk and chocolates [21] fast food and fried food [22]; snacks [21,23,28], high sugar, high fat [22], snack and diet [33]. The healthy pattern was called healthy [21], traditional in the country [23,24,26,30], vegetables, meat [25], Mediterranean [26], meat [25], Portuguese [26], Fruit-Milk-Eggs; Animal organs and refined cereals [35]. Each number represents an association found in the articles between all dietary patterns and associated factors. Associations are demonstrated directly or inversely, with patterns that were classified as healthy or unhealthy. In each study, it was possible to find more than one association between dietary pattern and associated factor.

## Data Availability

Not applicable.

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
