# Peer review of "Factors Associated with Dietary Patterns of Schoolchildren: A Systematic Review"

_nutrients, 2023, doi:10.3390/nu15112450_

Round 1

Reviewer 1 Report

Dear Authors,

I am grateful for the opportunity to review your manuscript entitled „Factors associated with dietary patterns of schoolchildren: a systematic review“. The manuscript is well-written, and it was a pleasure to review it.

After reviewing the manuscript, I have a couple of suggestions.

In lines 41-42, "The evaluation of food consumption in childhood is essential to help understand the effect of food choices on health". Did you think about the food choices that can affect health outcomes in adulthood? It may be helpful to rephrase the sentence to improve reading comprehension.

Methods:

Additionally, could the authors provide a complete search strategy in the methods or in the supplementary material to ensure the research can be reproduced in the future?

In line 80, when mentioning "influencing factors" e "influence," did you intend to use the word "and" instead of „e“ between?

Placement of references within the text starting from line 97 to the end of the text (manuscript)

I noticed that you placed references in the middle of sentences and directly after the author's presentation of research. It is recommended to place references at the end of sentences instead, as they refer to the entire sentence and the cited research.

In line 133, please place references in square brackets.

Limitations lines 317-319.

It seems that the authors were discussing how using a different method to gather data on eating habits might impact the findings of this research. Can Authors furnish further details on the impact of using a different methodology to gather information on dietary patterns on the findings of the study.?

References

It is necessary to format references according to MDPI instructions

Author Response

Dear Editors and Reviewers of Nutrients,

Subject: Response addressing the reviewer's comments about the article entitled: “Factors associated with dietary patterns of schoolchildren: a systematic review” – Manuscript ID: nutrients-2386468

Dear Editors,

We appreciate your willingness to consider our article for possible publication. We respectfully present the clarifications requested by the reviewers as well as the manuscript changes that was made to better address each of their suggestions. Thus, we hope to meet the expectations of this renowned journal.

Reviewer 1

Dear Authors,

I am grateful for the opportunity to review your manuscript entitled ¨Factors associated with dietary patterns of schoolchildren: a systematic review“. The manuscript is well-written, and it was a pleasure to review it.

After reviewing the manuscript, I have a couple of suggestions.

 1) In lines 41-42, "The evaluation of food consumption in childhood is essential to help understand the effect of food choices on health". Did you think about the food choices that can affect health outcomes in adulthood? It may be helpful to rephrase the sentence to improve reading comprehension.

Answer:

Changes were made to the second paragraph of the "Introduction" section. The assessment of food consumption in childhood is essential to help to understand the effect of food choices on health in the short term and even during adult life, as eating habits acquired in childhood can remain in the long term.

2) Methods: Additionally, could the authors provide a complete search strategy in the methods or in the supplementary material to ensure the research can be reproduced in the future?

Answer:

Thanks for the suggestion, the search strategy was attached in the supplementary material.

3) In line 80, when mentioning "influencing factors" e "influence," did you intend to use the word "and" instead of "e" between?

Answer:

Yes, that was a translation error. The passage was rewritten as suggested.

4) Placement of references within the text starting from line 97 to the end of the text (manuscript)

Answer:

All references had been carefully reviewed as suggested.

5) I noticed that you placed references in the middle of sentences and directly after the author's presentation of research. It is recommended to place references at the end of sentences instead, as they refer to the entire sentence and the cited research.

Answer:

References were organized as recommended.

6) In line 133, please place references in square brackets.

Answer:

The change was made as suggested.

7) Limitations lines 317-319. It seems that the authors were discussing how using a different method to gather data on eating habits might impact the findings of this research. Can Authors furnish further details on the impact of using a different methodology to gather information on dietary patterns on the findings of the study?

Answer:

Assessing food consumption is essential to characterize and evaluate the dietary patterns of a given population. However there are limitations, inherent to different dietary intake assessment (quantitative or semiquantitative food frequency questionnaire, 24-hour recalls, 3-day food daily record, etc), that may impair the knowledge of the foods and/or nutrients consumed, such as difficulty with the concept of time, short attention, memory and reduced vocabulary to know the components of the diet and preparations, excess or omission in estimating the sizes of portions of food consumed, and others, which can lead to different results from dietary patterns.  Most of the articles included used the FFQ whose longest period of coverage is far from the 24-hour recall, which measures current dietary intake. Despite these, it was decided to include different methods of dietary assessment due to the lack of consensus and diversity of results for reflection.

8) References: It is necessary to format references according to MDPI instructions

Answer:

All references have been carefully reviewed as suggested.

All points placed were carefully reviewed and we hope to meet the journal’s expectations. In addition, a rereading and adaptation of all topics requested by the reviewers was carried out.

Regards,

Laura Rangel Drummond de Menezes

 Rafaela Cristina Vieira e Souza

Pollyanna Costa Cardoso

Luana Caroline dos Santos

Reviewer 2 Report

Materials and Methods are well described. The systematic review has been correctly set although it is quite surprising that only 16 papers were finally eligible. 

I would suggest to organized Table 1 in geographical zones because that will help the understanding of the patterns. Nevertheless, some studies (Liao et al., Lee et al, etc) do not really classify a pattern but some food categories that are highly consumed. Authors should justify if these lists of food categories could be considered as patterns and if so how could you compare them to what other authors set as patterns.

Table 2 is not clear. What do the numbers mean?. Number of associations or factors or papers?

Could lines 293-297 be supported by additional cross references.

What about ethnicity?. Have authors found out any significant conclusion?. Why did authors skip mentioning the countries where these studies have been conducted?. 

Are there differences between the children and adolescent studies?. Maybe the tables should also refer or organize the results by age groups.

Line 328 and before, authors mention unhealthy foods but could it be better to refer to unhealthy habits?. What it is unhealthy is the consumption pattern of these foods like snacks etc.

Lines 331-332: are these statements conclusions of this review?. I would say no.  Suggesting  "it is important to stimulate free play and the practice of physical activity by providing accessible and safe urban spaces" might be a great recommendation for some pathologies but maybe not for food patterns that is what the review is about. 

What about family interventions?. What about recommendations to industry or regulators?. What about labelling information?.

Conclusions need to be changed.

Author Response

Dear Editors and Reviewers of Nutrients,

Subject: Response addressing the reviewer's comments about the article entitled: “Factors associated with dietary patterns of schoolchildren: a systematic review” – Manuscript ID: nutrients-2386468

Dear Editors,

We appreciate your willingness to consider our article for possible publication. We respectfully present the clarifications requested by the reviewers as well as the manuscript changes that was made to better address each of their suggestions. Thus, we hope to meet the expectations of this renowned journal.

Reviewer 2

1) Materials and Methods are well described. The systematic review has been correctly set although it is quite surprising that only 16 papers were finally eligible.

Answer:

The quality of a systematic review is influenced by its planning and the rigor of its stages, in addition to the quality of the included studies. To start the systematic review, we determined a guiding phrase: “How is the eating pattern of schoolchildren around the world and what are its associated factors?”. So, we identified the databases to be consulted, defined the search strategies, built the PECOS strategy and the inclusion and exclusion criteria, applied the criteria in the selection of articles, evaluate critically the individual quality of the articles (using the Newcastttle Ottawa Scale) made by 3 evaluators and a fourth if it needs. It is convenient to add that the population has a lack of studies in the literature on the subject, being one of the initial research interests.

2) I would suggest to organized Table 1 in geographical zones because that will help the understanding of the patterns. Nevertheless, some studies (Liao et al., Lee et al, etc) do not really classify a pattern but some food categories that are highly consumed. Authors should justify if these lists of food categories could be considered as patterns and if so how could you compare them to what other authors set as patterns.

Answer:

Data from the country of where each study was carried out was included in Table 1, as suggested. The classification of patterns for our study is essential since it is the main theme and we understand that all articles included dietary patterns. To make such a choice, one of the criteria used was the investigation of the statistical method chosen in each article, in all of them principal component analysis or cluster analysis was used, thus allowing each of them to be included in the review after being read full text.

3) Table 2 is not clear. What do the numbers mean?. Number of associations or factors or papers?

Answer:

Table 2 represents the number of associations found in the articles, between all dietary patterns and associated factors found in each study in a broad but objective way. Associations are demonstrated directly or inversely, with patterns that were classified as healthy or unhealthy. In each study it is possible to find more than one association between dietary pattern and associated factor. This information was inserted in the document to present the table in the results, as suggested.

4) Could lines 293-297 be supported by additional cross references.

Answer:

References have been added, as suggested.

5) What about ethnicity?. Have authors found out any significant conclusion?. Why did authors skip mentioning the countries where these studies have been conducted?.

Answer:

Only one article (Kaiser et al. 2015) presented ethnicity data related to the mother (the mother's level of acculturation was positively related to the consumption of fast and convenience foods). Countries were included in table 1, as suggested by another reviewer.

6) Are there differences between the children and adolescent studies?. Maybe the tables should also refer or organize the results by age groups.

Answer:

Studies that covered both children and adolescents (such as Li et al. 2016, Galvan Portillo et al. 2018, Zamora Gasga et al. 2017, Silva et al. 2012, Shrof et al. 2013, O'Brien et al. 2013, Rahmawaty et al., 2014) associated socioeconomic factors, parental education, nutritional status and screen time with unhealthy and healthy patterns, results obtained similar to studies with a restricted children's audience. These articles did not distinguish the results according to the age group of interest.

It should be noted that the focus of this review refers to schoolchildren and that the inclusion of adolescents occurred only because this cycle of life is encompassed. The articles included, as shown in Table 2 contains the age range of the studies when it comes to the sample (3rd column). For a better visualization of ages, as suggested, studies from the same country were placed in the table in ascending order of age. (China, Mexico and Brazil).

7) Line 328 and before, authors mention unhealthy foods but could it be better to refer to unhealthy habits?. What it is unhealthy is the consumption pattern of these foods like snacks etc.

Answer:

Thank you for the suggestion, it was relevant and we changed in the manuscript.

8) Lines 331-332: are these statements conclusions of this review?. I would say no.  Suggesting  "it is important to stimulate free play and the practice of physical activity by providing accessible and safe urban spaces" might be a great recommendation for some pathologies but maybe not for food patterns that is what the review is about.

Answer:

The “conclusion” section has been changed as suggested. It was highlighted that the care and attention of health professionals are necessary in the promotion and monitoring of inadequate eating habits in childhood, as early nutritional intervention can favor the prevention of diseases in the near and long future.

9) What about family interventions?. What about recommendations to industry or regulators?. What about labelling information?.

Answer:

Given these findings, it is essential that public policies aimed at food and nutrition education actions in schools and in the community are implemented and intensified.

In addition, it is essential to ensure healthier environments that protect from the exposure of ultra-processed food marketing, to regulate the commercialization of these foods, to propose regulatory actions for advertisements and to encourage the consumption of in natura foods and to inhibit free access to ultra-processed foods, facilitating the understanding of labels and overcoming barriers to promoting healthy eating.

The “conclusion” section was changed to clarify these points.

10) Conclusions need to be changed.

Answer:

The “conclusion” section was changed, as suggested.

All points placed were carefully reviewed and we hope to meet the journal’s expectations. In addition, a rereading and adaptation of all topics requested by the reviewers was carried out.

Regards,

Laura Rangel Drummond de Menezes

 Rafaela Cristina Vieira e Souza

Pollyanna Costa Cardoso

Luana Caroline dos Santos

Reviewer 3 Report

This is a study that uses a systematic review to examine the dietary patterns of schoolchildren. Overall, the topic is very interesting, and the writing is quite fluent. However, I have the following concerns, and I hope the author can make adjustments and improvements based on these key points.

1. In the abstract, the author mentions the inclusion of studies that examine the income and dietary patterns of schoolchildren aged 7-10 years old. However, Table 1 includes many articles that do not fall within this age range, such as the Kaiser 2015 study that examines children as young as 2 years old, and the Galvan-Portillo 2018 study that examines children as old as 15 years old. Additionally, the age range of participants in the Li 2016 study is not clearly stated. Therefore, I recommend that the author revise this section to ensure consistency across the abstract, inclusion criteria, and literature table.

2. Table 1 is the essence of this systematic review, but the author's current organization lacks a results column. Therefore, I recommend adding this column to Table 1. Additionally, although the author clearly lists the dietary pattern classifications of each study, it is not clear which ones are the "healthy" or "unhealthy" foods that are most important to this study. Please provide further clarification in this regard. After expanding Table 1, it may be beneficial to consider a horizontal layout to improve the table's readability.

3. Table 2 requires reformatting, as currently, each English word is split into several lines, making it difficult to read and visually unappealing. Additionally, it is not clear what the numbers 1, 2, and 3 represent in the table, as the author does not specify their meaning. This represents a deficiency in the presentation of research methods and results that can be improved.

4. Regarding the classification of "healthy" and "unhealthy" foods, it is unclear whether the author classified them independently or based on the definitions provided by each individual study. Food classification can be challenging, as demonstrated by the example of Liao 2022, which categorizes animal organs and refined cereals together. It is unclear whether this grouping is considered healthy or unhealthy since red meat is typically considered a good prototype food, whereas refined cereals are often considered unhealthy. Additionally, both Garcia-Chavez 2020 and Vieira 2019 classify bread as "traditional," but it is a processed starch and should likely be classified as unhealthy. In contrast, Lee 2017 classifies bread as a snack.

5. In real-life scenarios, most meals are a mixture of different types of foods, such as having a boiled egg with toast and jam, along with chocolate milk for breakfast. Would this meal be considered unhealthy because of the chocolate milk, or would it be considered healthy because it contains a boiled egg? In a systematic review, we would expect the author to have a clearer definition or classification of foods to address these questions when reading individual studies. This way, when the author claims that 93.75% of the foods examined are unhealthy, there will be a more solid foundation for the statement.

6. There are several details in English writing that require attention. I recommend that the author have a proficient English writer or copyeditor review the entire manuscript for these issues. For instance, in the abstract, it is best to avoid starting a sentence with a number, as seen in line 21. Additionally, in line 69, the PROSPERO notation is incorrect and requires a reference citation and hyperlink. In line 70, the author misspells the full name of PRISMA. It is unclear what "e" means in line 80. Finally, in line 87, the author should ensure that the punctuation is correct when using "observational - cross-sectional." These are just a few examples, and I suggest the author thoroughly reviews and checks the entire manuscript.

7. In the discussion section, the author did not provide enough comparison with existing literature. Several review articles in this field are commonly cited, yet the author did not reference or mention them, which is unfortunate. I recommend that the author explains the contributions of these articles and their unique findings compared to their study. If the articles are highly relevant, they should be added to the discussion. If they are not relevant, the author should explain why their team considers them not relevant.

Scaglioni S, De Cosmi V, Ciappolino V, Parazzini F, Brambilla P, Agostoni C. Factors Influencing Children's Eating Behaviours. Nutrients. 2018 May 31;10(6):706. doi: 10.3390/nu10060706.

Sadeghirad B, Duhaney T, Motaghipisheh S, Campbell NR, Johnston BC. Influence of unhealthy food and beverage marketing on children's dietary intake and preference: a systematic review and meta-analysis of randomized trials. Obes Rev. 2016 Oct;17(10):945-59. doi: 10.1111/obr.12445. Epub 2016 Jul 18.

Rocha NP, Milagres LC, Longo GZ, Ribeiro AQ, Novaes JF. Association between dietary pattern and cardiometabolic risk in children and adolescents: a systematic review. J Pediatr (Rio J). 2017 May-Jun;93(3):214-222. doi: 10.1016/j.jped.2017.01.002. Epub 2017 Feb 23.

Hill DC, Moss RH, Sykes-Muskett B, Conner M, O'Connor DB. Stress and eating behaviors in children and adolescents: Systematic review and meta-analysis. Appetite. 2018 Apr 1;123:14-22. doi: 10.1016/j.appet.2017.11.109. Epub 2017 Dec 2.

As mentioned above.

Author Response

Dear Editors and Reviewers of Nutrients,

Subject: Response addressing the reviewer's comments about the article entitled: “Factors associated with dietary patterns of schoolchildren: a systematic review” – Manuscript ID: nutrients-2386468

Dear Editors,

We appreciate your willingness to consider our article for possible publication. We respectfully present the clarifications requested by the reviewers as well as the manuscript changes that was made to better address each of their suggestions. Thus, we hope to meet the expectations of this renowned journal.

Reviewer 3

This is a study that uses a systematic review to examine the dietary patterns of schoolchildren. Overall, the topic is very interesting, and the writing is quite fluent. However, I have the following concerns, and I hope the author can make adjustments and improvements based on these key points.

  1. In the abstract, the author mentions the inclusion of studies that examine the income and dietary patterns of schoolchildren aged 7-10 years old. However, Table 1 includes many articles that do not fall within this age range, such as the Kaiser 2015 study that examines children as young as 2 years old, and the Galvan-Portillo 2018 study that examines children as old as 15 years old. Additionally, the age range of participants in the Li 2016 study is not clearly stated. Therefore, I recommend that the author revise this section to ensure consistency across the abstract, inclusion criteria, and literature table.

Answer:

The target age group of our study, the schoolchildren, comprises the age group from 07 to 10 years old, but a portion of the evaluated studies extended beyond this age (as Kaiser et al. 2015 that directed the target audience in two phases: 2 to 4 years old and 4 to 8 years old, Galvan-Portilho et al. 2018: 5 to 15 years old, Li et al. 2016: 7 to 15 years old, Rahmawaty et al. 2014: 9 to 13 years old, Shroff et al. 2013: 5 to 12 years old, Zamora-Gasga et al. 2017: 9 to 12 years old and Garcia-Chaves et al. 2020: 5 to 11 years old), as explained in the characterization of the studies. The summary was changed to clarify age range coverage, as suggested.

2.Table 1 is the essence of this systematic review, but the author's current organization lacks a results column. Therefore, I recommend adding this column to Table 1. Additionally, although the author clearly lists the dietary pattern classifications of each study, it is not clear which ones are the "healthy" or "unhealthy" foods that are most important to this study. Please provide further clarification in this regard. After expanding Table 1, it may be beneficial to consider a horizontal layout to improve the table's readability.

Answer:

A column with this information was inserted on Table 1 as suggested. In addition, Table 1 was revised following the reviewers' suggestions, including horizontal formatting.

For this determination between healthy and unhealthy foods carried out in the study, between healthy and unhealthy, the classification of foods was used: ultra-processed (industrial formulation made from substances derived from food or synthesized in laboratories and consumption should be avoided), processed (manufactured by the industry with added salt or sugar and are direct derivatives of in natura foods and should be consumed in moderation) and in natura and minimally processed (obtained directly from plants or animals without having undergone any alteration after leaving nature or only processes cleaning tools and should be eaten liberally), which categorizes foods by degree of processing. Thus, in their constitution as described in the methods, unhealthy patterns are determined by: “snacks (n=4), “junk convenient” (n=2), "unhealthy” (n=1)," less healthy” (n=1), "industrialized” (n=1)," saturated fat” (n=1), “junk food” (n=1), “fast food and fried food" (n=1), "obesogenic" (n=1), "snacks" (n=1), "soft drinks" (n=1), "canned soups and meals" (n=1), "sweetened drinks" (n=1), "modern" (n=1), “milk and chocolates" (n=1), "high fat" (n=1), "high sugar" (n=1), "Western" (n=1), “diet" (n=1), “refined cereals and animal organs” (n=1).

Additional reference: Brasil. National Health (2014) Guia alimentar para a população brasileira, 2 ed. Brasília, 154 p.

  1. Table 2 requires reformatting, as currently, each English word is split into several lines, making it difficult to read and visually unappealing. Additionally, it is not clear what the numbers 1, 2, and 3 represent in the table, as the author does not specify their meaning. This represents a deficiency in the presentation of research methods and results that can be improved.

We added a footnote to clarify these points. In addition, we have reformulated Table 2, as suggested, for better visualization.

“Each number represents an association found in the articles between all dietary patterns and associated factors. Associations are demonstrated directly or inversely, with patterns that were classified as healthy or unhealthy. In each study it is possible to find more than one association between dietary pattern and associated factor.”

  1. Regarding the classification of "healthy" and "unhealthy" foods, it is unclear whether the author classified them independently or based on the definitions provided by each individual study. Food classification can be challenging, as demonstrated by the example of Liao 2022, which categorizes animal organs and refined cereals together. It is unclear whether this grouping is considered healthy or unhealthy since red meat is typically considered a good prototype food, whereas refined cereals are often considered unhealthy. Additionally, both Garcia-Chavez 2020 and Vieira 2019 classify bread as "traditional," but it is a processed starch and should likely be classified as unhealthy. In contrast, Lee 2017 classifies bread as a snack.

Answer:

The classification of a pattern is a challenge because it reflects the social, cultural, religious and environmental characteristics of a people. For this classification in this study, between healthy and unhealthy, the categorization of foods by the degree of processing was used - ultra-processed, processed and in natura and minimally processed, as understood in question 2. Therefore, for the classification, the following criteria was used: if more than half of the foods were in the ultraprocessed category or if the study had already called them unhealthy, they were classified as "unhealthy". The patterns that mostly contained in natura or minimally healthy foods or called themselves healthy were classified as “healthy”. The detailed explanation is included in the last topic of the "Methods" section.

Additional reference: Brasil. National Health (2014) Guia alimentar para a população brasileira, 2 ed. Brasília, 154 p.

  1. In real-life scenarios, most meals are a mixture of different types of foods, such as having a boiled egg with toast and jam, along with chocolate milk for breakfast. Would this meal be considered unhealthy because of the chocolate milk, or would it be considered healthy because it contains a boiled egg? In a systematic review, we would expect the author to have a clearer definition or classification of foods to address these questions when reading individual studies. This way, when the author claims that 93.75% of the foods examined are unhealthy, there will be a more solid foundation for the statement.

Answer:

This critical view of dietary patterns should provide us with a broad view of the patterns, that is, when we observe that 93.75% of the studies found an unhealthy pattern (which is mostly characterized by unhealthy foods) among the analyzed patterns, we can, therefore, to verify that in our study population, worldwide, the consumption of nutritionally unbalanced foods was associated with deleterious health effects, regardless of gender or income. This result is an indication that there is a trend in the food consumption of the studied population (unhealthy or ultra-processed foods), worldwide, and that there are adverse associations with health.

We added this information in text as suggested in the subtopic “General characteristics of dietary patterns of the included studies”.

  1. There are several details in English writing that require attention. I recommend that the author have a proficient English writer or copyeditor review the entire manuscript for these issues. For instance, in the abstract, it is best to avoid starting a sentence with a number, as seen in line 21. Additionally, in line 69, the PROSPERO notation is incorrect and requires a reference citation and hyperlink. In line 70, the author misspells the full name of PRISMA. It is unclear what "e" means in line 80. Finally, in line 87, the author should ensure that the punctuation is correct when using "observational - cross-sectional." These are just a few examples, and I suggest the author thoroughly reviews and checks the entire manuscript.

Answer:

The suggested changes were made in the “Methods” section. All manuscript writing was proofread by a translator.

  1. In the discussion section, the author did not provide enough comparison with existing literature. Several review articles in this field are commonly cited, yet the author did not reference or mention them, which is unfortunate. I recommend that the author explains the contributions of these articles and their unique findings compared to their study. If the articles are highly relevant, they should be added to the discussion. If they are not relevant, the author should explain why their team considers them not relevant.

Scaglioni S, De Cosmi V, Ciappolino V, Parazzini F, Brambilla P, Agostoni C. Factors Influencing Children's Eating Behaviours. Nutrients. 2018 May 31;10(6):706. doi: 10.3390/nu10060706.

Sadeghirad B, Duhaney T, Motaghipisheh S, Campbell NR, Johnston BC. Influence of unhealthy food and beverage marketing on children's dietary intake and preference: a systematic review and meta-analysis of randomized trials. Obes Rev. 2016 Oct;17(10):945-59. doi: 10.1111/obr.12445. Epub 2016 Jul 18.

Rocha NP, Milagres LC, Longo GZ, Ribeiro AQ, Novaes JF. Association between dietary pattern and cardiometabolic risk in children and adolescents: a systematic review. J Pediatr (Rio J). 2017 May-Jun;93(3):214-222. doi: 10.1016/j.jped.2017.01.002. Epub 2017 Feb 23.

Hill DC, Moss RH, Sykes-Muskett B, Conner M, O'Connor DB. Stress and eating behaviors in children and adolescents: Systematic review and meta-analysis. Appetite. 2018 Apr 1;123:14-22. doi: 10.1016/j.appet.2017.11.109. Epub 2017 Dec 2.

Answer:

The Hill DC study, Moss RH, Sykes-Muskett B, Conner M, O'Connor DB. Stress and eating behaviors in children and adolescents: Systematic review and meta-analysis. Appetite. 2018 Apr 1;123:14-22. doi: 10.1016/j.appet.2017.11.109. Epub 2017 Dec 2.

The article above was not considered coherent by the team. The study deals with the effect of stress on eating behavior. It should be noted that some factors, although hypothesized and related to the health of the student, were not investigated because they detract from the proposed objective. In addition, association by dietary intake and not objectively dietary patterns was investigated in the review.

The other references were added as suggested.

All points placed were carefully reviewed and we hope to meet the journal’s expectations. In addition, a rereading and adaptation of all topics requested by the reviewers was carried out.

Regards,

Laura Rangel Drummond de Menezes

 Rafaela Cristina Vieira e Souza

Pollyanna Costa Cardoso

Luana Caroline dos Santos

Round 2

Reviewer 1 Report

Dear authors, 

I want to express my gratitude for accepting the suggestions made. The manuscript has been further enhanced.

Reviewer 2 Report

Answers and changes look fine.

The manuscript has been improved according to the 1st review suggestions.

Reviewer 3 Report

The authors have made adjustments and responses to all the issues very carefully, and they have handled them well within the existing framework. I can accept most of the responses, except for one final point. Please clarify this point for both me and future readers of the paper.

Regarding Table 2, the authors state that 1/2/3 represent associations, which can be either direct or inverse and can indicate either healthy or unhealthy. However, it is unclear what 1 represents, what 2 means, and what 3 represents. Could the authors please provide a direct explanation? Typically, associations are represented using a correlation coefficient, which ranges from -1 to +1.